# Ancient origin of lubricated joints in bony vertebrates

Amjad Askary[1,2†], Joanna Smeeton[1,2†], Sandeep Paul[1,2], Simone Schindler[1,2], Ingo Braasch[3,4,5], Nicholas A Ellis[6], John Postlethwait[3], Craig T Miller[6], J Gage Crump[1,2*]

[1]Department of Stem Cell Biology and Regenerative Medicine, Keck School of Medicine of University of Southern California, Los Angeles, United States; [2]Eli and Edythe Broad CIRM Center for Regenerative Medicine and Stem Cell Research, Keck School of Medicine of University of Southern California, Los Angeles, United States; [3]Institute of Neuroscience, University of Oregon, Eugene, United States; [4]Department of Integrative Biology and Program in Ecology, Michigan State University, East Lansing, United States; [5]Department of Evolutionary Biology and Behavior, Michigan State University, East Lansing, United States; [6]Department of Molecular and Cell Biology, University of California, Berkeley, Berkeley, United States

**Abstract** Synovial joints are the lubricated connections between the bones of our body that are commonly affected in arthritis. It is assumed that synovial joints first evolved as vertebrates came to land, with ray-finned fishes lacking lubricated joints. Here, we examine the expression and function of a critical lubricating protein of mammalian synovial joints, Prg4/Lubricin, in diverse ray-finned fishes. We find that *Prg4* homologs are specifically enriched at the jaw and pectoral fin joints of zebrafish, stickleback, and gar, with genetic deletion of the zebrafish *prg4b* gene resulting in the same age-related degeneration of joints as seen in lubricin-deficient mice and humans. Our data support lubricated synovial joints evolving much earlier than currently accepted, at least in the common ancestor of all bony vertebrates. Establishment of the first arthritis model in the highly regenerative zebrafish will offer unique opportunities to understand the aetiology and possible treatment of synovial joint disease.

*For correspondence: gcrump@usc.edu

†These authors contributed equally to this work

Competing interests: The authors declare that no competing interests exist.

## Introduction

Synovial joints allow for free movement between adjacent bones and are characterized by a fluid-filled cavity separating layers of hyaline articular cartilage. The synovial cavity is enclosed by a membrane, which is often strengthened externally by a fibrous capsule and contains lubricating molecules, such as hyaluronic acid and lubricin, that reduce friction at the joint surface (*Koyama et al., 2014*; *Rhee et al., 2005*). A prevailing hypothesis is that lubricated synovial joints first evolved in tetrapods in response to newfound mechanical challenges imposed on the weight-bearing joints of nascent limbs (*van der Kraan, 2013a*, *2013b*) (*Figure 1A*). Whereas previous histological studies had suggested that the jaw joints of lungfish (a lobe-finned fish like humans) (*Bemis, 1986*), and potentially longnose gar and sturgeon (ray-finned fishes) (*Haines, 1942*), have synovial-like morphology, the extent to which these joints are molecularly and functionally similar to tetrapod synovial joints had remained untested. In particular, the assumption that ray-finned fishes lack the sophisticated types of lubricated joints found in humans has hampered the use of the zebrafish model for the study of synovial joint diseases such as arthritis. By examining the expression and function of

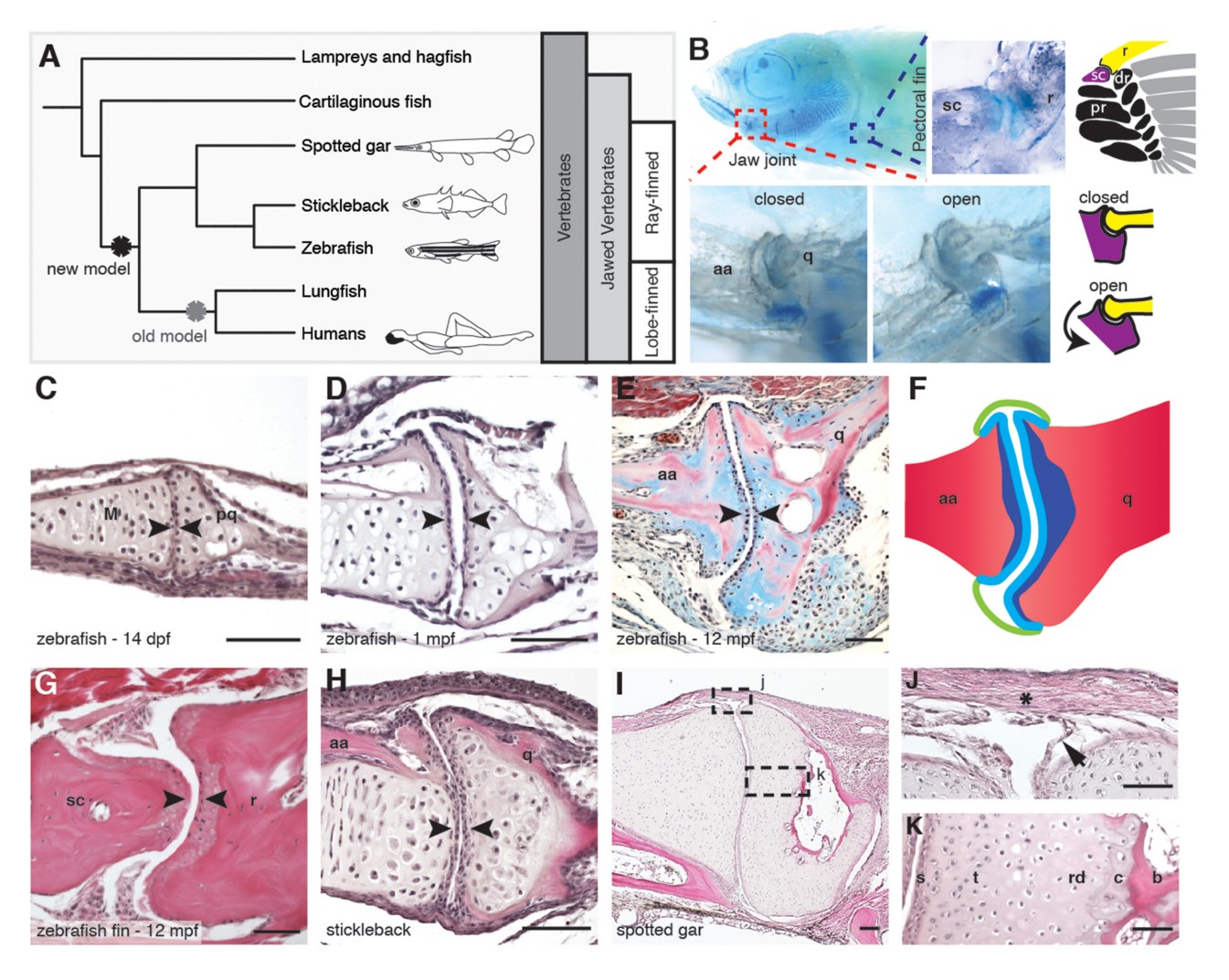

**Figure 1.** Synovial-like morphology of jaw and fin joints in ray-finned fish. (**A**) Phylogenetic tree contrasts the old model of synovial joint evolution (grey asterisk) with the new model of synovial joints evolving in a common precursor of all bony vertebrates (black asterisk). (**B**) Alcian Blue-stained adult zebrafish and accompanying diagrams show the pectoral fin joints, and jaw joints in open and closed positions. (**C–E, G–I**) Sections of 14 dpf (n = 4), 1 mpf (n = 3), and adult (n = 6) zebrafish jaw joints; ray-scapula joint in the adult zebrafish pectoral fin (n = 4); and stickleback (1 mpf, n = 3) and spotted gar (10.2 cm, n = 3) jaw joints. Sections are stained by H&E (**C, D, G–I**) or trichrome (**E**). Articular chondrocytes (black arrowheads) line the cavity. (**F**) Schematic of adult jaw joint shows bone (red), cartilage (blue, lighter shade indicates articular), and synovial membrane (green). (**J, K**) Magnifications of (**I**) show the synovial membrane (arrow), fibrous capsule (asterisk) and multilayered articular cartilage (**K**). Scale bar in h, 100 μm; all other panels, 50 μm. aa: anguloarticular; q: quadrate; sc: scapula; r: ray; pr: proximal radial; dr: distal radial; M: Meckel's; pq: palatoquadrate; s: superficial; t: transitional; rd: radial layer; c: calcified cartilage; b: bone.

homologs of a critical lubricating protein of mammalian joints, Lubricin, we provide evidence that certain joints of adult zebrafish are indeed true synovial joints.

## Results

### Synovial-like morphology of the jaw and fin joints of ray-finned fishes

Given the suggested synovial-like morphology of jaw joints in several fishes, we examined whether joints of the widely used teleost species, the zebrafish (*Danio rerio*), also display synovial morphology (*Figure 1A*). Bone μCT of adult zebrafish shows that the jaw joint, an articulation between the anguloarticular and quadrate bones, resembles a hinge joint (*Video 1*), with manual opening and

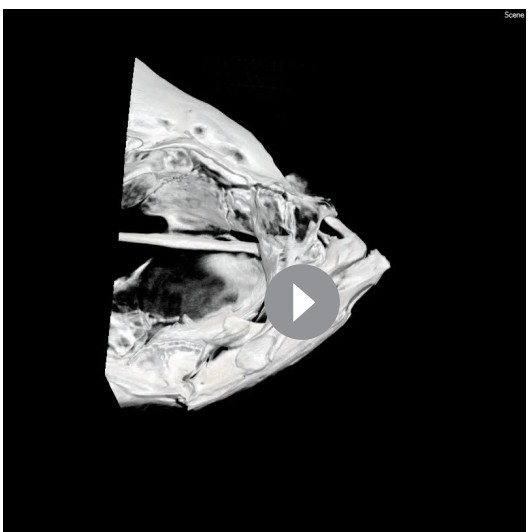

**Video 1.** Bone μCT of the adult zebrafish head. 3D reconstruction of bone shows the structure and relative position of the jaw joint within the adult zebrafish head (12 mpf). Color-coding at the end of the movie illustrates the anguloarticular (pink) and quadrate (yellow) bones that articulate at the jaw joint.

closing of the mouth in fixed Alcian-Blue-stained animals revealing movement in a single plane (*Figure 1B*). While study of zebrafish has contributed to our understanding of the embryonic development of the jaw joint, zebrafish larvae at the most commonly studied stage, 6 days post-fertilization (dpf), show little evidence of a synovial cavity (*Miller et al., 2003*). However, whether this joint acquires synovial characteristics later had not been described. Our histological investigations revealed the variable presence of a partial cavity as early as 14 dpf, and a prominent and consistently present cavity by 28 dpf (*Figure 1C,D*). In adult zebrafish (12 months post-fertilization, mpf), distinct layers of flattened articular chondrocytes line the jaw joint cavity, with hypertrophic chondrocytes located beneath the articular surface (*Figure 1E,F*). We also observed similar cavities lined by flattened chondrocytes in joints of the pectoral fin in adult zebrafish, in particular between the proximal and distal radials and between the marginal ray and scapula (*Figure 1B,G*). We next used *sox10* and *tricho-rhino-phalangeal syndrome 1 (trps1)* transgenes (*Askary et al., 2015*) to visualize jaw joint cavitation over time in living zebrafish. At 3, 7, and 14 dpf, the early chondrocyte marker

*trps1*:GFP labels the subset of *sox10*:dsRed+ chondrocytes at the joint surface, with a partial cavity variably apparent at 14 dpf (*Figure 2A–C*). At 1, 2 and 8 mpf, *trps1*:GFP is maintained in articular chondrocytes as the cavity expands, with a subset of cells marked by *sox10*:dsRed (*Figure 2D–F*). In the jaw of the relatively small zebrafish, the presence of a cavity and joint-lining cells, with a minimal fibrous capsule, is consistent with it being a synovial-like joint in miniature, similar to the homologous incudomalleolar synovial joint of the mammalian middle ear (*Whyte et al., 2002*).

To examine whether synovial-like morphology is a conserved feature of ray-finned fish, we also examined juveniles of the distantly related teleost fish, the three-spined stickleback (*Gasterosteus aculeatus*), and an outgroup of teleost fish, the ray-finned spotted gar (*Lepisosteus oculatus*) (*Braasch et al., 2016*). Both stickleback and spotted gar jaw joints displayed synovial cavities lined by flattened cells (*Figure 1H,I*). In the jaw joint of the larger spotted gar (standard length 10.2 cm), an internal one-cell-thick membrane and thick external fibrous capsule enclosed the cavity, with joint cartilage divided into the same superficial, transitional, radial, and calcified layers seen in mammalian synovial joints (*Figure 1J,K*). The presence of these additional morphological features in the gar jaw supports synovial-like features being an ancestral property of bony vertebrates.

## Expression of *Prg4* homologs at joints of diverse fishes

Given the synovial-like morphology of several types of joints in fish, we next examined whether the chondrocytes lining these joints share a common molecular signature with those of mammalian joints. Chondrocytes lining mammalian synovial joints differ from those in the growth plate by expressing *Prg4*, which encodes a lubricin proteoglycan that forms a cross-linked network with hyaluronan and Aggrecan to reduce friction across the joint surface (*Jay and Waller, 2014*). Consistent with the appearance of a synovial cavity in juvenile stages, we find that the joint-lining cells of the zebrafish jaw express *prg4b* starting from 15 dpf and continuing throughout adulthood (*Figure 3A–D*). We observed much weaker levels of expression at other joints of the face, such as the midline ceratohyal-ceratohyal joint (*Figure 3A*, arrowhead) and hyoid joint (*Figure 3E*), which lack synovial morphology. Expression of *prg4b* was not detected in the jaw joint at earlier stages (7 dpf, data not shown), consistent with the late onset of *Prg4* expression at mammalian joints (*Rhee et al., 2005*). In addition, *prg4b* expression appeared in joint-lining cells of the synovial-like ray-scapula articulation

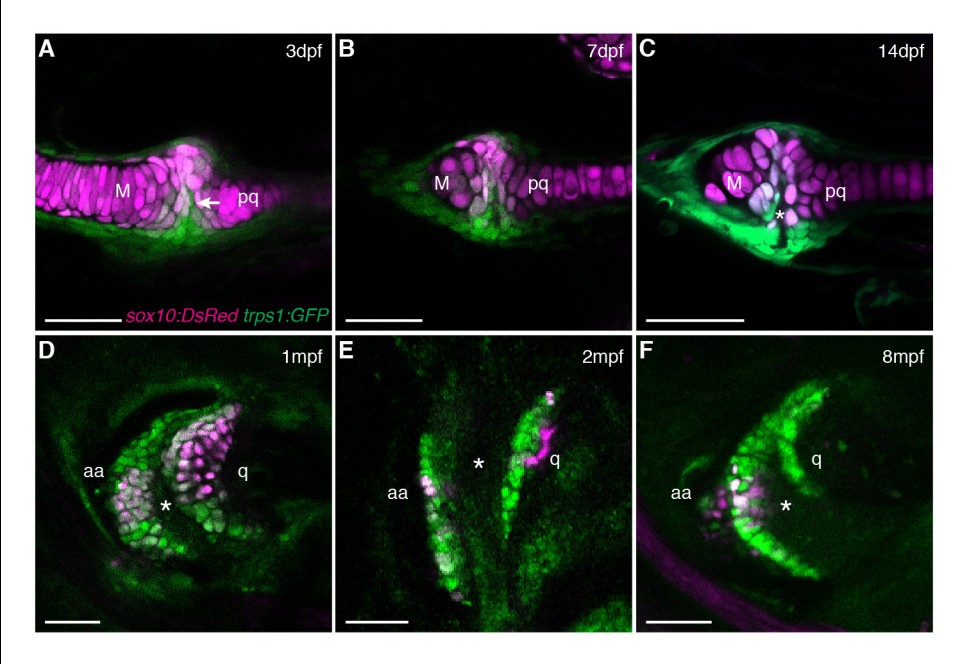

**Figure 2.** Live imaging of jaw joint cavitation. (**A–F**) In double transgenic zebrafish, *sox10*:DsRed marks all chondrocytes and *trps1*:GFP labels nascent joint chondrocytes (arrow) and perichondrial cells at 3 and 7 dpf. By 14 dpf, a partial cavity (asterisk) was evident in 1/6 animals. In 3/3 animals at 1 mpf, 4/4 at 2 mpf, and 4/4 at 8 mpf, a fully formed cavity is evident and *sox10*:DsRed is expressed in a subset of *trps1*:GFP+ articular chondrocytes. Scale bar, 50 μm.

of the pectoral fin at 3 mpf, but not in the non-synovial intervertebral discs (*Figure 3F,G*). Zebrafish also expressed *prg4b* outside of joints, including conserved expression with mammalian *Prg4* in liver (*Ikegawa et al., 2000*) and possibly ligaments (*Sun et al., 2006*) (*Figure 3A,G*). Similar to zebrafish, stickleback expressed *prg4b* and gar expressed *prg4* in joint-lining cells of the juvenile jaw (*Figure 3H–J*). In contrast, the related *prg4a* gene is not enriched at the jaw joint in either zebrafish or stickleback, instead showing expression throughout cartilage (*Figure 3—figure supplement 1*). Of note, the lineage leading to the spotted gar diverged before the teleost genome duplication (*Amores et al., 2011*), resulting in zebrafish and stickleback having two *Prg4* co-orthologs and gar a single ortholog (*Figure 3—figure supplement 2*). Analysis of the single *prg4* gene in gar therefore reveals that enriched expression of *Prg4* within articular chondrocytes existed before the divergence of ray-finned and lobe-finned vertebrates.

We next examined whether articular chondrocytes of the zebrafish jaw display other features of chondrocytes lining mammalian synovial joints, including enriched expression of hyaluronan synthase (Has) enzymes in the radial layer (*Hiscock et al., 2000*) and lower levels of types II and X Collagen, Aggrecan, and Matrilin compared to growth plate chondrocytes (*Khan et al., 2007*). Consistently, we observed relative depletion of *col10a1*, *acana*, and *matrilin1* mRNA and Col2a1a and Aggrecan protein in articular versus deeper chondrocytes, as well as enriched expression of *has3* in the radial layer of the juvenile zebrafish jaw joint (*Figure 3K–M*, and *Figure 3—figure supplement 1*). Although none of these markers are exclusive to synovial joints, the shared expression of Lubricin and hyaluronan synthase enzymes and relative absence of cartilage maturation genes (e.g. Collagen II/X, Aggrecan, Matrilin1) demonstrates a common molecular signature between articular chondrocytes of the zebrafish jaw and mammalian synovial joints.

## Requirement of zebrafish *prg4b* for adult maintenance of joints

A major function of the synovial cavity is to lubricate the joint, with loss of lubrication resulting in age-related joint degeneration. We therefore asked whether expression of *Prg4* orthologs by cells lining synovial-like joints in ray-finned fishes reflects a conserved requirement of lubricin protein in

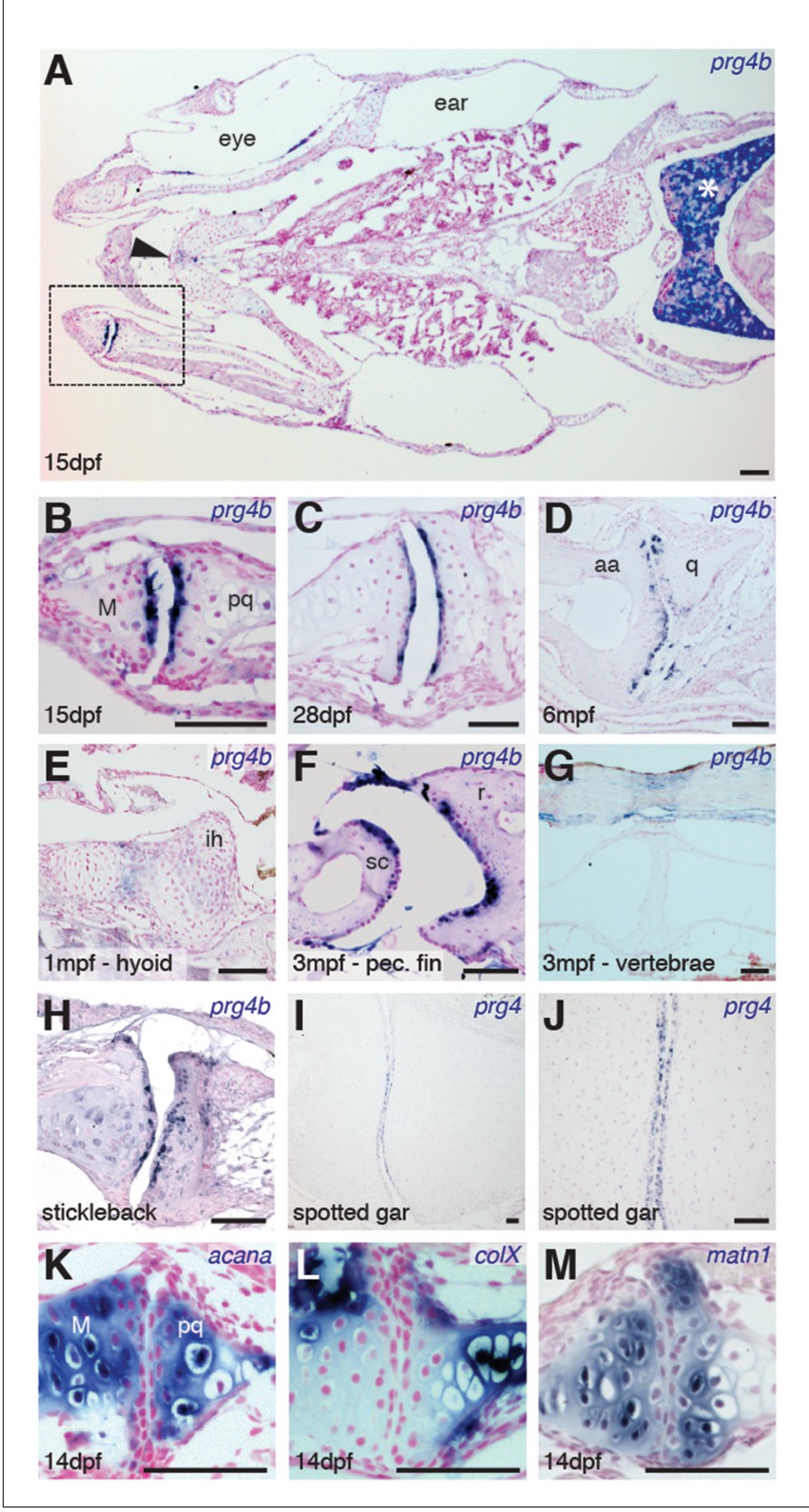

**Figure 3.** Expression of Prg4 genes in articular chondrocytes of ray-finned fish. (A–G) *prg4b* expression in articular chondrocytes of the zebrafish jaw joint (boxed region in **A**, **B–D**), hyoid joint (**E**); ray-scapula joint (**F**); and
*Figure 3 continued on next page*

*Figure 3 continued*

vertebral column (**G**). *prg4b* is also expressed in the liver (asterisk), possibly in ligaments above the vertebrae, and weakly at the ceratohyal-ceratohyal joint (arrowhead). *n* = 3 each. (**H–J**) Expression of stickleback *prg4b* (1 mpf, *n* = 3) and gar *prg4* (10.2 cm, *n* = 3) in jaw joint articular chondrocytes (**J**, magnification of **I**). (**K–M**) Exclusion of *acana*, *col10a1*, and *matn1* expression from articular chondrocytes of the zebrafish jaw. *n* = 3 each. Scale bar, 50 μm. ih: interhyal. See also *Figure 3—figure supplement 1* and *2*.

The following figure supplements are available for figure 3:

**Figure supplement 1.** Gene expression within the zebrafish and stickleback jaw joints.

**Figure supplement 2.** Evolution of vertebrate Proteoglycan 4 (Lubricin).

lubricating and hence maintaining these joints. To do so, we used TALE nucleases (*Huang et al., 2011*; *Sander et al., 2011*) to generate loss-of-function deletion alleles for zebrafish *prg4a* and *prg4b* (*Figure 4A*). Mice lacking *Prg4* function (*Koyama et al., 2014*; *Rhee et al., 2005*) and humans with homozygous loss of *PRG4* in Camptodactyly-arthropathy-coxa vara-pericarditis syndrome (*Marcelino et al., 1999*) have a progressive deterioration of joint surfaces that includes loss of articular chondrocytes, accumulation of acellular matrix in the cavity, synovial hyperplasia, and thickening of the deep chondrocyte layer. Consistent with the lack of early joint defects in *Prg4-/-* mice, zebrafish doubly mutant for *prg4a* and *prg4b* had no defects in the jaw joint at either 6 dpf or 1 mpf and no gross defects of the adult skeleton (*Figure 4B–D*). However, *prg4a-/-; prg4b-/-* zebrafish began to display a weak accumulation of acellular matrix in limited domains of the joint surface at 2 mpf (*Figure 4E*), a phenotype that became more severe by 6 mpf (*Figure 4F*). By 12 mpf, *prg4a-/-; prg4b-/-* zebrafish had multiple jaw joint abnormalities, including an acellular matrix at the joint surface, synovial hyperplasia, increased numbers of deep chondrocytes, and in some cases complete erosion of the joint surface accompanied by underlying bone defects (*Figure 4G*, *Figure 4—figure supplement 1*, and additional examples in *Figure 4—figure supplement 2*). Quantification of jaw joint defects using an Osteoarthritis Research Society International (OARSI) scoring system (*Pritzker et al., 2006*) that we modified for zebrafish (*Figure 4—figure supplement 2*) confirmed that joint defects increased in severity during aging (*Figure 4H*), consistent with the progressive arthritis seen in *Prg4-/-* mice (*Koyama et al., 2014*; *Rhee et al., 2005*) and patients with CACP (*Marcelino et al., 1999*). Defects found in *prg4a-/-; prg4b-/-* double mutants were also not confined to the jaw joint, appearing in the ray-scapula and inter-radial joints of the pectoral fin (*Figure 5A–C*). Further, consistent with only *prg4b* being enriched in jaw joint chondrocytes, *prg4b* but not *prg4a* single mutants displayed ray-scapula fin joint defects and a similar severity of jaw joint defects as double mutants at 12 mpf (*Figure 4G,H* and *Figure 5B*). In contrast, we detected no changes in the hyoid joints of the face and the intervertebral discs in 12 mpf *prg4a-/-; prg4b-/-* mutants (*Figure 5D, E*), consistent with these joints lacking synovial cavities and high-level *prg4b* expression. Our data therefore support a specific requirement for zebrafish *Prg4* homologs in the adult maintenance of joints with synovial-like morphology.

## Discussion

Although we do not know if Prg4b protein is secreted into joint cavities in zebrafish, or whether the cavities are fully enclosed as in mammals, the finding that the jaw and fin joints of zebrafish require the molecular lubricant lubricin for their maintenance supports zebrafish having synovial-like lubricated joints. Contrary to existing dogma (*van der Kraan, 2013b*), our data suggest that lubricated synovial joints evolved before vertebrates moved onto land in at least the common ancestor to all bony vertebrates. Although fish are not subject to gravity in the same way as tetrapods, the evolution of synovial joints in fish-like ancestors may have facilitated movement of first the jaw, and then the fins, against water resistance. Interestingly, recent studies show that the radial joints of the pectoral fin, which we find to have synovial morphology and *prg4b* dependency in zebrafish, are homologous to those of the tetrapod wrist (*Gehrke et al., 2015*). The existence of these synovial joints at

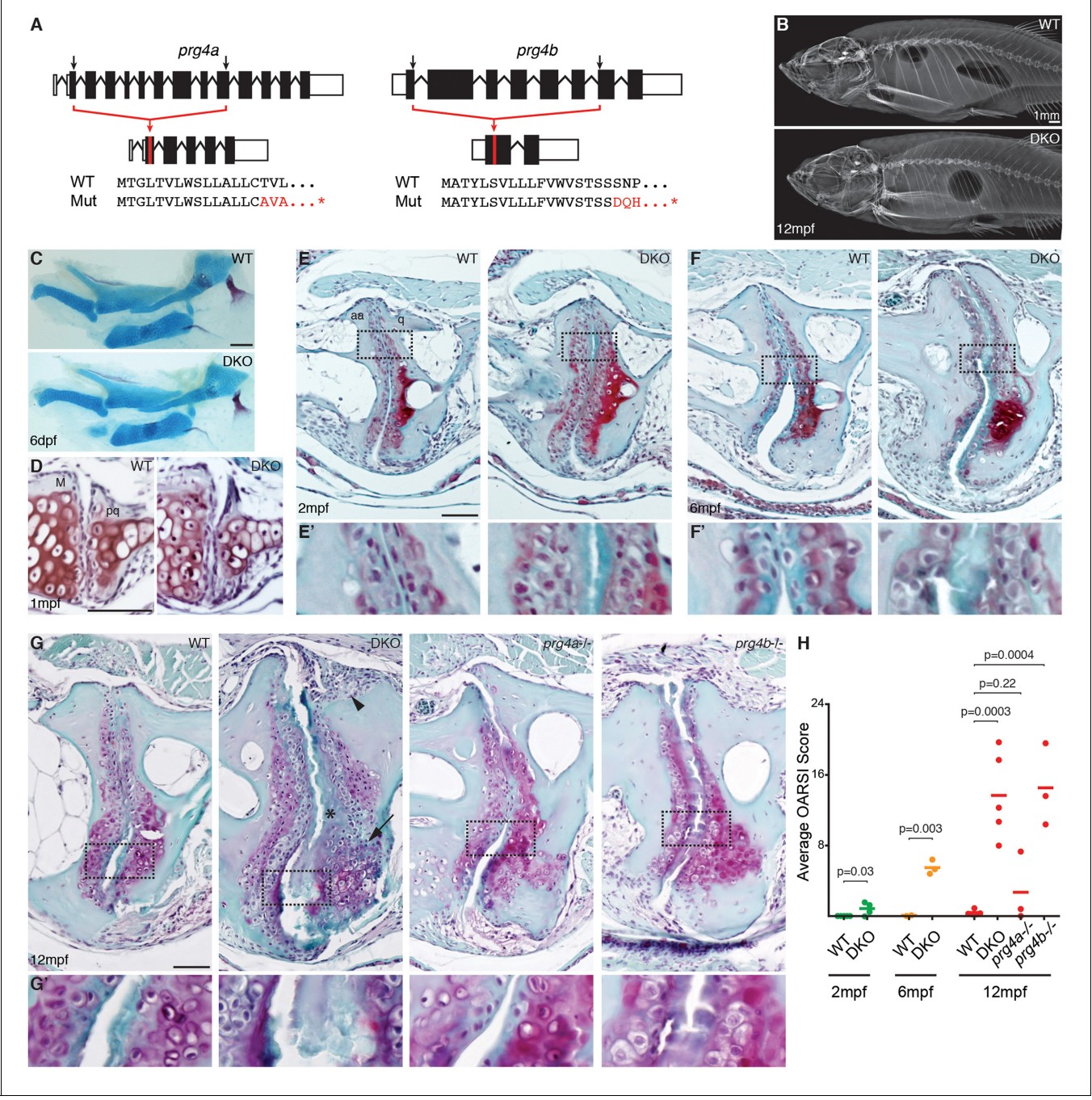

**Figure 4.** Progressive deterioration of the jaw joint in zebrafish lacking prg4b. (**A**) Schematics of *prg4a* and *prg4b* TALEN mutants show deleted sequences. (**B**) X-ray imaging shows no gross morphological bone defects of *prg4a-/-; prg4b-/-* mutants (DKO) at 12 mpf. (**C**) Alcian Blue staining shows normal facial cartilages in DKO at 6 dpf. (**D**) SafraninO staining shows a normal jaw joint between Meckel's (M) and palatoquadrate (pq) cartilages in DKO at 1 mpf. (**E–G**) SafraninO staining at 2, 6 and 12 mpf shows increasingly abnormal joints between the jaw anguloarticular (aa) and quadrate (q) bones in DKO. Defects include acellular matrix accumulation in the cavity (asterisks, see magnified regions in **E'–G'**) at all stages, and synovial hyperplasia (arrowheads) and expanded deep chondrocytes (arrows) at 12 mpf. Single *prg4b* but not *prg4a* mutants showed similar jaw joint defects to DKO at 12 mpf. (**H**) Quantification of jaw joint defects using our modified OARSI system for zebrafish. Scale bar 50 μm, except in (**B**). See also *Figure 4—figure supplements 1,2*, and *Figure 4—source data 1*.

The following source data and figure supplements are available for figure 4:

**Source data 1.** Quantification of joint defects in zebrafish lacking *prg4* genes.

*Figure 4 continued on next page*

*Figure 4 continued*

**Figure supplement 1.** Serial sections through a representative wild-type and *prg4a-/-; prg4b-/-* mutant jaw joint.
**Figure supplement 2.** OARSI scoring system for zebrafish.

the base of the pectoral fins of ancestral bony fish may therefore have facilitated their later functional evolution into the larger synovial joints of tetrapod limbs. Our findings show that zebrafish can be a relevant model to understand the development of synovial specializations, including the poorly understood process of cavitation. The establishment of the first genetic model of arthritis in zebrafish will also allow a better understanding of the developmental progression of synovial joint disease. In the future, it will be exciting to test whether the articular cartilage lining synovial joints, which is affected in arthritis, displays the same regenerative potential as many of the other tissues in zebrafish (*Knopf et al., 2011*; *Morgan, 1900*; *Poss et al., 2002*).

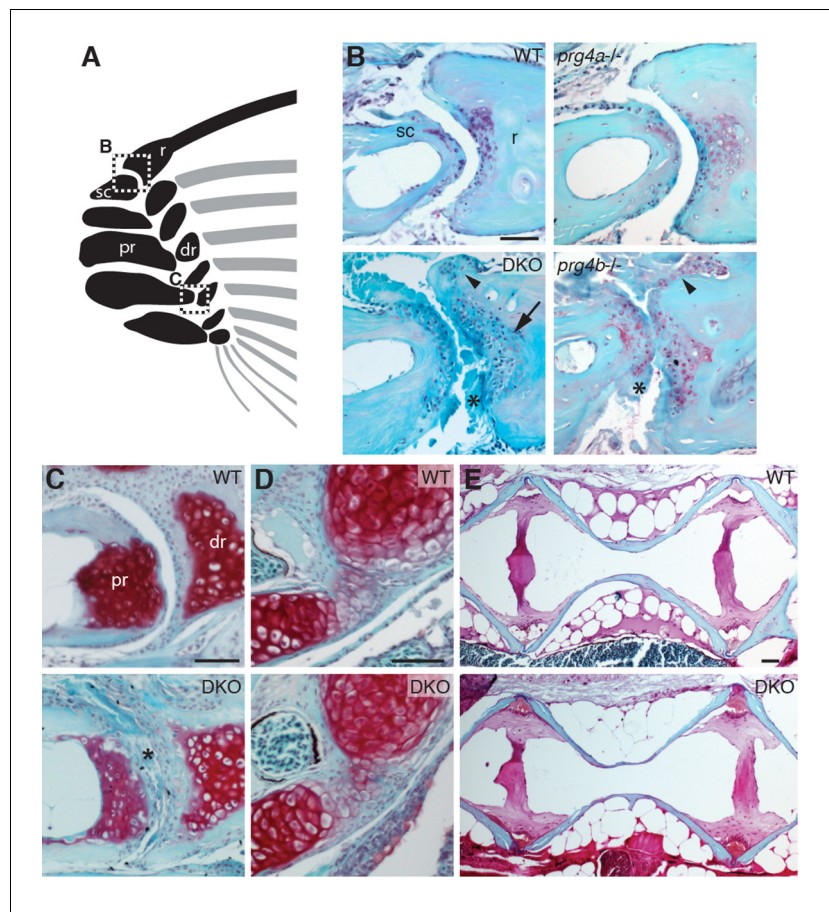

**Figure 5.** Requirement of Prg4 gene function for fin joints of zebrafish. (**A**) Schematic of pectoral fin joints. (**B–E**) SafraninO staining at 12 mpf shows abnormal joints between the ray (r) and scapula (sc) joint of the pectoral fin (**B**) in *prg4b* but not *prg4a* single mutants and *prg4a; prg4b* double mutants (DKO), and defects in the proximal radial (pr) and distal radial (dr) joint of the pectoral fin (**C**) in DKO. Defects include acellular matrix accumulation in the cavity (asterisks), synovial hyperplasia (arrowheads), and expanded deep chondrocytes (arrow). In contrast, the hyoid joint (**D**) and intervertebral discs (**E**) were normal in DKO. Phenotypes were consistently observed in three animals of each genotype. Scale bar 50 μm.

# Materials and methods

## Fish husbandry and zebrafish lines

The Institutional Animal Care and Use Committees of the University of Southern California, the University of California, Berkeley, and the University of Oregon approved all zebrafish (*Danio rerio*), three-spined stickleback (*Gasterosteus aculeatus*) and spotted gar (*Lepisosteus oculatus*) procedures, respectively – (IACUC #10885). Previously reported zebrafish lines used in this study include *Tg(sox10:dsRED)*[el10] (*Das and Crump, 2012*), *trps1*[j127aGt] (RRID:ZFIN_ZBD-GENO-100809-11) (*Talbot et al., 2010*), and *Tg(col2a1a_{BAC}:GFP)*[el483] (*Askary et al., 2015*).

## Phylogenetic tree

The phylogenetic tree was generated using phyloT (http://phylot.biobyte.de/) with NCBI taxonomy, and visualized in iTOL:Interactive Tree of Life (http://itol.embl.de/).

## Histology

Adult zebrafish samples were fixed in 4% PFA at 4°C for 7 days. Following fixation, animals were cut into smaller pieces to facilitate embedding and sectioning of the desired structures. The samples were then decalcified in 20% EDTA solution for 10 days at room temperature. For embedding, the tissue was first dehydrated through a series of ethanol washes (30, 50, 70, 95, and 100%) for 20 min each. Then ethanol was replaced with xylene substitute Hemo-De (Electron Microscopy Sciences, Hatfield, PA) in a series of 15 min washes (50, 75, and 100% Hemo-De). The samples were then incubated in a 1:1 Hemo-De:paraffin (Paraplast X-tra, VWR, Radnor, PA) solution at 65°C for an hour before an overnight incubation in 100% paraffin at 65°C. The following day, the samples were embedded in freshly melted paraffin. Larval and juvenile samples were prepared following the same general procedure with shorter fixation and decalcification steps (2 days fixation and 4 days decalcification). Animals younger than 3 weeks did not require decalcification. Juvenile stickleback and spotted gar samples were processed as above for adult zebrafish with the following changes: stickleback were decalcified in 20% EDTA for 7 days at room temperature; isolated spotted gar heads were fixed in 4% PFA at 4°C for 10–14 days and decalcified in 20% EDTA for 14 days at room temperature. For hematoxylin & eosin (H&E) staining, 5 µm paraffin sections were deparaffinized with xylene and re-hydrated through an ethanol series to distilled water. Sections were then stained in hematoxylin (VWR) for 2 min followed by brief acetic acid rinse and 2 min in Blueing Reagent solution (VWR); 2 × 30 s washes in water, 2 × 30 s in 95% ethanol. The sections were stained in Eosin (VWR) for 30 s followed by 3 × 1 min washes in 95% ethanol and 2 × 1 min washes in 100% ethanol. Following 2 × 2 min in Hemo-De, samples were mounted with cytoseal (Richard-Allan Scientific, Kalamazoo, MI) for imaging. For Trichrome staining, 5 µm paraffin sections were deparaffinized with xylene and re-hydrated through an ethanol series to distilled water. Trichrome stain was performed according to manufacturer's instructions using the Trichrome, Gomori One-Step, Aniline Blue Stain (Newcomer Supply, Middleton, WI). For SafraninO staining, 5 µm paraffin sections were deparaffinized with xylene and re-hydrated to distilled water. They were then stained in Weigert's Iron Hematoxylin (Newcomer Supply) for 5 min, washed in distilled water for 5 changes, differentiated in 0.06 N HCl solution in 70% ethanol for 2 s followed by 3 more washes in water. Sections were then stained in 0.02% Fast Green FCF (Sigma-Aldrich, St. Louis, MO) for 1 min and rinsed for 30 s in 1% acetic acid. Staining in 1% SafraninO (Newcomer Supply) was then performed for 30 min, followed by 3 × 1 min washes in 95% ethanol. Slides were then washed 2 × 1 min in 100% ethanol and 2 × 2 min in Hemo-De, before mounting with cytoseal for imaging. For Alcian staining, juvenile zebrafish were processed as described (*Walker and Kimmel, 2007*), and adult zebrafish were processed using a modified version of an online protocol (https://wiki.zfin.org/pages/viewpage.action?pageId=13107375).

## In situ hybridization

Probes were generated for zebrafish *prg4a, prg4b, has3, acana, col10a1*, and *matn1* using PCR amplification from cDNA. See *Supplementary file 1A* for primers used in amplification of zebrafish probe templates. Probes for stickleback *prg4a* and *prg4b* and gar *prg4* were synthesized from gBlocks Gene Fragments (Integrated DNA Technologies, Coralville, IA). The DNA sequences used for probe generation are listed in *Supplementary file 1B*. Stickleback *prg4b* sequence was

determined through sequence homology comparison with zebrafish Prg4b protein sequence using BlastP search in Ensembl. Two independent probes were designed for the putative stickleback *prg4b* (ENSGACG00000007505; ENSGACG00000007501) and both showed the same expression pattern in three independent animals. gBlocks or PCR products from cDNA amplification were subcloned into pCR-Blunt II TOPO vector (Invitrogen, Carlsbad, CA). Following sequence confirmation, digoxigenin (DIG)-labeled antisense probes were synthesized using T7 or SP6 RNA polymerase (Roche, Switzerland). In situ hybridization protocol was modified from (*Lien et al., 2006*). In brief, after de-paraffinization, slides were digested in 7.5 µg/ml proteinase K for 5 min and fixed in 4% PFA/0.2% gluteraldehyde for 20 min. Each slide was incubated overnight at 65°C with 1 µg of DIG-labeled riboprobe diluted in hybridization buffer. After hybridization, slides were washed three times in 1x SSC/Formamide at 65°C for 30 min, and three times in MABT for 15 min. Following 1 hr of blocking in 2% Blocking Buffer (Roche), hybridization was detected with anti-DIG-AP antibody (Roche, RRID:AB_514497) and developed with NBT/BCIP substrate colorimetric reaction (Roche). Slides were counterstained with nuclear fast red (Vector Laboratories, Burlingame, CA) prior to mounting.

## Immunofluorescence

Paraffin sections (5 µm) were de-paraffinized with xylene and rehydrated through an ethanol series to 1xPBST. Antigen retrieval was performed using pH 6.0 sodium citrate buffer in a steamer for 35 min. The sections were blocked with 2% donkey serum in PBST for 30 min at room temperature. The sections were incubated with primary antibodies against Col2a1 (goat polyclonal, SC7763, Santa Cruz Biotechnology, Santa Cruz, CA, RRID:AB_2229686) and Aggrecan (Cat# 13880-1-AP, Proteintech, Rosemont, IL) overnight at 4°C. Sections were further incubated in secondary antibodies and Hoechst 33342 nuclear stain for 1 hr at room temperature prior to mounting with Fluoromount-G (Southern Biotech, Birmingham, AL).

## Imaging

Fluorescence imaging of live animals and sections was performed using a Zeiss LSM5 confocal microscope. Histology and colorimetric in situ hybridization slides were photographed using a Leica D8 2500 microscope. Alcian Blue-stained samples were imaged using a Leica S8APO microscope. For microCT (µCT), adult zebrafish were euthanized and fixed in 4% PFA overnight and rinsed twice in 1x PBS. The fish head was dissected and glued to a Pasteur pipette to place it next to the scan head. The scans were performed in air on a XT H 225S T µCT scanner (Nikon Metrology, Brighton, MI) with a PerkinElmer 1621 detector at 120 kVp, 26 uA 500 ms exposure time resulting in an isotropic 3 micron voxel volume. A molybdenum target was used with no additional filtration of the beam. Raw data were reconstructed in CT Pro 3D v4.3.4 (XT Software Suite, Nikon Metrology) and video rendering performed on VG StudioMax v2.2 (Volume Graphics GmbH, Germany). X-ray imaging of fixed adult zebrafish utilized an UltraFocus60 x-ray cabinet (Faxitron Bioptics, Tucson, AZ).

## Construction of Prg4 knockout alleles

To make knockout alleles for *prg4a* and *prg4b*, two pairs of Transcription Activator-Like Effector Nucleases (TALENs) were used per gene to remove most of the coding sequence of the target genes. TALENs were designed and constructed as described previously (*Sanjana et al., 2012*). TALEN pairs were targeted toward the following sequences in the zebrafish genome: TTGGTCTC TTCTGGCTCTGC and TCGTCTGCTGCTCAGGGTGA for *prg4a* 5' TALENs, TAGGCGTCCCG TCACCCATT and CGCTGCAACTGCCAGGGCAA for *prg4a* 3' TALENs, TGCTGTTTGTGTGGGTC TCC and CACGTAAGCCAACAGATCGA for *prg4b* 5' TALENs, and TCCCCCAGCTGCAGCACTGG and TCTCACGAACCTGGAGAGGA for *prg4b* 3' TALENs. ARCA-capped RNA for each TALEN was transcribed in vitro using the mMESSAGE mMACHINE T7 Ultra Kit (Life Technologies, Carlsbad, CA). The RNA for four TALENs corresponding to each gene were mixed and injected into one-cell stage embryos to generate founder lines. Founders were then outcrossed to wild-type fish and the progenies were tested for genomic deletions. Deletion alleles were identified in the F1 generation by cloning and sequencing of amplicons that span the TALEN target sites for each gene. We obtained single *prg4a*[el687] (7686 bp deletion) and *prg4b*[el594] (11,638 bp deletion)

alleles by screening the progeny of 6 and 21 injected animals, respectively. For the list of genotyping primers see *Supplementary file 1C*.

## Zebrafish OARSI scoring system

5 μm serial sections throughout the jaw joint were stained with SafraninO and imaged as described above. For both the left and right jaw joint, three representative images were selected for each sample. Care was taken to select images that represent equivalent sectioning planes in wild-type and mutant joints. Grade (0–6) and stage (0–4) values were assigned to the anterior (anguloarticular) and posterior (quadrate) surfaces in each of the six joint images for each sample. The grade and stage value for each individual articular surface were multiplied to generate the score for that joint surface. Scores for both surfaces of the left and right joints were then averaged for each animal.

## Statistical analysis

Data are presented as a scatter plot with a line for the mean. Statistical analysis of average OARSI scores was performed using GraphPad Prism 7 (RRID:SCR_002798), with a two-tailed Student's t-test used to generate the p values.

## Acknowledgements

We thank Megan Matsutani and Jennifer DeKoeyer Crump for fish care and Seth Ruffins for video editing.

## Additional information

### Funding

| Funder | Grant reference number | Author |
| --- | --- | --- |
| National Institute on Deafness and Other Communication Disorders | T32 DC009975 | Amjad Askary |
| California Institute for Regenerative Medicine | Training Fellowship | Joanna Smeeton |
| National Institutes of Health | R01OD011116 | John Postlethwait |
| California Institute of Regenerative Medicine | New Faculty Award | J Gage Crump |
| Wright Foundation | Young Faculty Award | J Gage Crump |

The funders had no role in study design, data collection and interpretation, or the decision to submit the work for publication.

### Author contributions

AA, JS, Designed and performed the experiments, Analyzed the data and wrote the manuscript, Acquisition of data, Critically revised and approved the manuscript; SP, SS, Performed initial histology, Acquisition of data, Critically revised and approved the manuscript; IB, Contributed gar specimens, Analysis and interpretation of data, Contributed unpublished essential data or reagents, Critically revised and approved the manuscript; NAE, CTM, Contributed stickleback specimens, Contributed unpublished essential data or reagents, Critically revised and approved the manuscript; JP, Contributed gar specimens, Contributed unpublished essential data or reagents, Critically revised and approved the manuscript; JGC, Analyzed the data and wrote the manuscript, Conception and design, Critically revised and approved the manuscript

### Author ORCIDs

J Gage Crump, http://orcid.org/0000-0002-3209-0026

### Ethics

Animal experimentation: This study was performed in strict accordance with the recommendations in the Guide for the Care and Use of Laboratory Animals of the National Institutes of Health. The Institutional Animal Care and Use Committees of the University of Southern California, the University of California, Berkeley, and the University of Oregon approved all zebrafish (Danio rerio), three-spined stickleback (Gasterosteus aculeatus) and spotted gar (Lepisosteus oculatus) procedures, respectively. Protocol #10885 was approved by the Committee on the Ethics of Animal Experiments of the University of Southern California.

## Additional files

### Supplementary files

• Supplementary file 1. DNA sequences for in situ hybridization probes and genotyping. (1A) Forward and reverse primers used to amplify zebrafish cDNA for in situ hybridization probe generation. (1B) Sequences used to generate in situ hybridization probes for stickleback and spotted gar *prg4* genes. (1C) Forward and reverse primers used to genotype *prg4a* and *prg4b* mutant zebrafish.

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
