## [Decision Letter]

Thank you for submitting your article "Ancient origin of lubricated joints in bony vertebrates" for consideration by *eLife*. Your article has been favorably evaluated by Janet Rossant (Senior editor) and two reviewers, one of whom, Didier Stainier, is a member of our Board of Reviewing Editors.

The reviewers have discussed the reviews with one another and the Reviewing Editor has drafted this decision to help you prepare a revised submission.

Summary:

This concise and interesting paper from the Crump lab provides convincing evidence that synovial joints are present in ray-finned fishes, i.e., earlier than previously thought. Given the importance of joint dysfunction (including arthritis), this finding is significant, especially as the authors go on to develop a zebrafish model of arthritis.

Essential revisions:

1) The authors claim that their knockout model resembles osteoarthritis, which is a progressive disease. Therefore, adding quantitative data about the progression of the disease in these fish would reinforce the proposed model. For example, OARSI score can be provided for different time points.

2) Some of the figures, such as Figure 3, would be clearer and more informative by the addition of magnifications. Another example is Figure 4, where it isn't clear whether *matrilin1* is excluded from articular chondrocytes, as in higher vertebrates.

---

## [Author Response]

Essential revisions:

1) The authors claim that their knockout model resembles osteoarthritis, which is a progressive disease. Therefore, adding quantitative data about the progression of the disease in these fish would reinforce the proposed model. For example, OARSI score can be provided for different time points.

We have now examined jaw joint defects at two additional intermediate stages (2 and 6 months – new Figure 4). We also developed a modified OARSI scoring system for zebrafish (new Figure 4—figure supplement 2). This allowed us to generate quantitative data (new Figure 4) showing progressively worse joint defects from 2 to 6 to 12 months (in combination with our previous result of no joint defects at one month, Figure 4). This is now described in the main text: “Quantification of jaw joint defects using an Osteoarthritis Research Society International (OARSI) scoring system that we modified for zebrafish (Figure 4—figure supplement 2) confirmed that joint defects increased in severity during aging (Figure 4), consistent with the progressive arthritis seen in *Prg4-/-* mice and CACP patients.”

*2) Some of the figures, such as Figure 3, would be clearer and more informative by the addition of magnifications. Another example is Figure 4, where it isn't clear whether matrilin1 is excluded from articular chondrocytes, as in higher vertebrates.*

For the OA phenotypes in Figure 4 (i.e. old Figure 4 and new 2mpf and 6mpf sections), we now provide magnified boxes to highlight changes at the joint surface. We have also repeated the matrilin1 in situ. Compared to the original version where the in situ reaction was overexposed, this new image (Figure 3) more clearly shows exclusion of matrilin1 from the surface layer of articular chondrocytes at 14 dpf. We also now show continued exclusion of matrilin1 from articular chondrocytes of the jaw joint at 1 month (Figure 3—figure supplement 1).